# Conditional RNA interference in mammalian cells via RNA transactivation

Yu Zhou[1,2,8], Peike Sheng [1,3,4,8], Jiayi Li[5,6], Yudan Li [6,7], Mingyi Xie [1,3,4] ✉ & Alexander A. Green [5,6,7] ✉

RNA interference (RNAi) is a powerful tool for sequence-specific gene knockdown in therapeutic and research applications. However, spatiotemporal control of RNAi is required to decrease nonspecific targeting, potential toxicity, and allow targeting of essential genes. Herein we describe a class of de-novo-designed RNA switches that enable sequence-specific regulation of RNAi in mammalian cells. Using cis-repressing RNA elements, we engineer RNA devices that only initiate microRNA biogenesis when binding with cognate trigger RNAs. We demonstrate that this conditional RNAi system, termed **O**rthogonal **R**NA **I**nterfer**en**ce induced by **T**rigger **R**NA (ORIENTR), provides up to 14-fold increases in artificial miRNA biogenesis upon activation in orthogonal libraries. We show that integration of ORIENTR triggers with dCas13d enhances dynamic range to up to 31-fold. We further demonstrate that ORIENTR can be applied to detect endogenous RNA signals and to conditionally knockdown endogenous genes, thus enabling regulatory possibilities including cell-type-specific RNAi and rewiring of transcriptional networks via RNA profile.

RNAi has been actively studied over the years and has proven to be a powerful tool for target gene knockdown in both fundamental biological research and medical applications[1]. RNAi can be achieved through delivered siRNA (small interfering RNA)[2] or miRNAs (microRNAs)[3], selectively silencing target gene expression through post-transcriptional regulation without perturbing the genome. However, constitutively active, unregulated RNAi suffers from several drawbacks that limit the range of potential applications. First, delivered siRNA competes with the endogenous RNAi machinery, potentially diverting cellular resources in detrimental ways[4]. Second, systemic delivery of unregulated siRNA or miRNA has spatial limitations, and RNAi activity in nontargeted cells or tissues can lead to undesirable off-target effects[5]. Third, RNAi that lacks temporal regulation is limited in specific applications that require inducible gene knockdown to avoid lethal effects when targeting essential genes or to

control the onset and progression of desired phenotypes[6–9]. In the last decade, researchers have sought to control RNAi activity in a time- or locus-specific manner for conditional reactivation by various molecular inducers, including small-molecule riboswitches[10–12], modified antisense oligonucleotides[13] and small synthetic RNAs[14,15].

RNA switch technology holds great promise in transducing an RNA input signal to an RNAi output signal as a means of precisely regulating the activity of a gene of interest. The high programmability and structural predictability of RNA enables automated de novo design for constructing sensing elements and logic circuits with the potential to be interfaced with endogenous biological pathways[16–18]. In recent years, RNA switch techniques have been successfully employed in mammalian cells to upregulate transgene activity. For instance, eToeholds achieved RNA-specific transgene translational activation by regulating RNA structure at internal ribosome entry sites (IRESs)[19].

[1]UF Center for NeuroGenetics (CNG), Gainesville, FL, USA. [2]Department of Molecular Genetics and Microbiology (MGM), University of Florida, Gainesville, FL, USA. [3]Department of Biochemistry and Molecular Biology, College of Medicine (COM), University of Florida, Gainesville, FL, USA. [4]UF Health Cancer Center, Gainesville, FL, USA. [5]Department of Biomedical Engineering, Boston University, Boston, MA, USA. [6]Biological Design Center, Boston University, Boston, MA, USA. [7]Molecular Biology, Cell Biology and Biochemistry Program, Boston University, Boston, MA, USA. [8]These authors contributed equally: Yu Zhou, Peike Sheng. ✉e-mail: mingyi.xie@ufl.edu; aagreen@bu.edu

Several studies have established programmable RNA-sensing-dependent protein translation technologies in which specific RNA base-pairing motifs recruit ADARs to edit RNA for transgene translational activation[20–22]. However, the downregulation of endogenous genes in mammalian cells through programmable RNA switch techniques remains largely unexplored despite many potential applications where conditional gene silencing is desired[23].

With the goal of developing conditional RNAi that can be regulated by cellular RNAs in mammalian cells, we carefully investigated the possibility of perturbing molecular recognition between RNA substrate and essential enzymes in the RNAi pathway. In the canonical pathway, miRNA biogenesis is initiated in the nucleus by the Microprocessor complex, which is comprised of the nuclear RNase III Drosha and two DGCR8 proteins as cofactors. Microprocessor searches through the RNA transcripts to identify and cleave primary (pri-)miRNAs, releasing hairpin-shaped precursor (pre-)miRNAs that are then exported into the cytoplasm for further cleavage by Dicer into mature miRNA[24,25]. Both Microprocessor and Dicer recognize substrate RNAs by conserved RNA motifs and specific secondary structures, thus providing the opportunity to regulate substrate RNA accessibility by disruption of Microprocessor or Dicer recognition elements[26–28]. Prior efforts to conditionally regulate RNAi activity have focused on Dicer substrate RNA formation[14,15]; however, Dicer substrate RNA requires characteristic double-stranded RNA helical ends with 5′ phosphate and a 3′ 2-nt overhang[27,28]. These structural requirements make it challenging to incorporate cis-regulatory elements for substrate activation. Thus far, Dicer substrate-based RNA transducing systems have only employed trans-acting RNA molecules in test tubes[14] and cell lysate[15] and have yet to be demonstrated in live cells.

Herein we report a conditional RNAi system that functions robustly in mammalian cells and responds to specific RNA stimuli by harnessing cis-regulatory RNA elements to control recognition by the Microprocessor. The system, which we term **O**rthogonal **R**NA **I**nterfer**en**ce induced by **T**rigger **R**NA (ORIENTR), consists of a conditional pri-miRNA that requires binding of a cognate trigger RNA to form an active Microprocessor substrate that can, in turn, be processed into an artificial miRNA (amiRNA) for induced target gene suppression. In the absence of the trigger RNA, the conditional pri-miRNA adopts an inactive secondary structure that precludes Microprocessor recognition. We explore the sequence and structural requirements for a functional pri-miRNA scaffold and identify functional orthogonal ORIENTR devices in human cells. We further harness a deactivated CRISPR nuclease dCas13d to protect trigger RNA from degradation and to increase RNA nuclear localization for improved performance. Importantly, ORIENTRs completely decouple the trigger RNA sequence in the sensor region from the output amiRNA sequence in the actuator region such that an arbitrary RNA input can be used to silence any desired mRNA. Indeed, we are able to use ORIENTR to sense fluctuations in endogenous mRNA expression under environmental stress and to conditionally trigger amiRNA biogenesis targeting endogenous genes. Our approach thus demonstrates an important step for establishing programmable conditional RNAi synthetic circuits in mammalian cells.

## Results

### Investigation of functional pri-miRNA requirements

To develop ORIENTRs, we first investigated the impact of different cis-regulatory RNA elements on recognition by the Microprocessor. Previous studies[24,27] indicate that the typical Microprocessor substrate RNA consists of an apical loop, a ~22-bp stem with guide RNA and passenger RNA on each side, an imperfect 11-bp basal stem as a molecular ruler to direct Drosha cleavage, and flanking single-stranded RNA on both sides. Aside from these structural requirements, certain conserved sequence motifs are also required for efficient and homogeneous enzyme processing, including the UG motif at the basal

junction, UGU/GUG motif in the apical loop, the mismatched GHG (mGHG) motif in basal stem, and CNNC in the 3′ flanking sequence[26,29] (Fig. 1a). Although cleavage efficiency and homogeneity are correlated with the number of these cis-elements in the substrate, it is believed that a pri-miRNA with a stable basal stem plus only one sequence motif is adequate to define a Microprocessor substrate[30]. Based on the above molecular features, we identified the basal stem of the pri-miRNA, which adopts a conserved structure with relatively flexible sequence requirements, as an ideal target to regulate substrate accessibility through structural manipulations.

We chose pri-miR-16-2 as the Microprocessor substrate scaffold to study experimentally as it possesses signature pri-miRNA structural motifs and a well-resolved crystal structure in complex with Microprocessor has been published[31]. By swapping the original miR16-2 sequence with a viral miRNA sequence (miR-HSUR4)[32], the pri-miRNA was transcribed from a pol III promoter (U6 promoter) and repurposed to silence a green fluorescent protein (GFP) reporter harboring a miR-HSUR4 target site to directly monitor RNAi activity (Supplementary Fig. S1a and see Supplementary Table S1 for sequences). We further confirmed that the knockdown efficiency from the pri-miR-16-2 scaffold is independent of the promoter and is comparatively more potent than other scaffolds[33,34] using a luciferase reporter or a GFP reporter as the target gene (Supplementary Fig. S1b, c and see Supplementary Table S2 for sequences). To investigate the design flexibility of an effective pri-miRNA scaffold, we first modified the upstream sequence flanking the scaffold basal stem by modifying its sequence, structure, or both its sequence and structure (Supplementary Fig. S1d and see Supplementary Table S3 for sequences). All three altered scaffolds retained their effectiveness in silencing GFP, demonstrating that the upstream sequence of the pri-miRNA basal stem does not possess essential RNA motifs for a functional pri-miRNA scaffold and that upstream hairpin structure does not sterically hinder downstream amiRNA biogenesis. The sequence/structure flexibility can facilitate the integration of cis-regulatory RNA motifs for our later conditional pri-miRNA designs.

Next, to investigate the structural and sequence constraints for pri-miRNA, we swapped the native pri-miR 16-2 basal stem region (m and m′*, where′* designates an imperfect reverse complement) with scrambled control sequences (n and n′*) that preserved the original basal stem structure with randomized sequences designed by NUPACK (Fig. 1b, Supplementary Fig. S1e and see Supplementary Table S4 for sequences). As expected, when either m or m′* was replaced by n or n′* to disrupt the basal stem, the scaffold failed to knock down reporter GFP. However, when both sequences were replaced while preserving the structure, the pri-miRNA scaffold recovered its potency in repressing the GFP (Fig. 1b). Our hypothesis that a functional pri-miR-16-2 scaffold is dependent on the basal stem structure without sequence constraints was further confirmed by northern blot (Supplementary Fig. S1f). Based on these results, we reasoned that the mGHG and UG motifs are dispensable for a functional pri-miRNA in our system and that the basal stem requires a conserved structure instead of a conserved sequence, providing flexibility in conditional pri-miRNA design. Our results agree with the previous structural findings that in pri-miR-16-2, the mGHG motif interacts with the Microprocessor through structural features without apparent sequence preference[31].

### ORIENTR design and performance

Based on the above results, we devised a design for ORIENTRs that built upon earlier work on aptamer-based RNA switches used for in vitro diagnostics[18,35]. In our ORIENTR design, the 11-nt sequence in the 5′ half of the basal stem was sequestered in a hairpin structure to preclude the correct pri-miRNA substrate structure from folding (blue region, Fig. 1c). Disruption of basal stem folding prevents Drosha acquisition and processing of the substrate pri-miRNA. This upstream hairpin can be opened by interacting with a 37-nt cognate RNA trigger

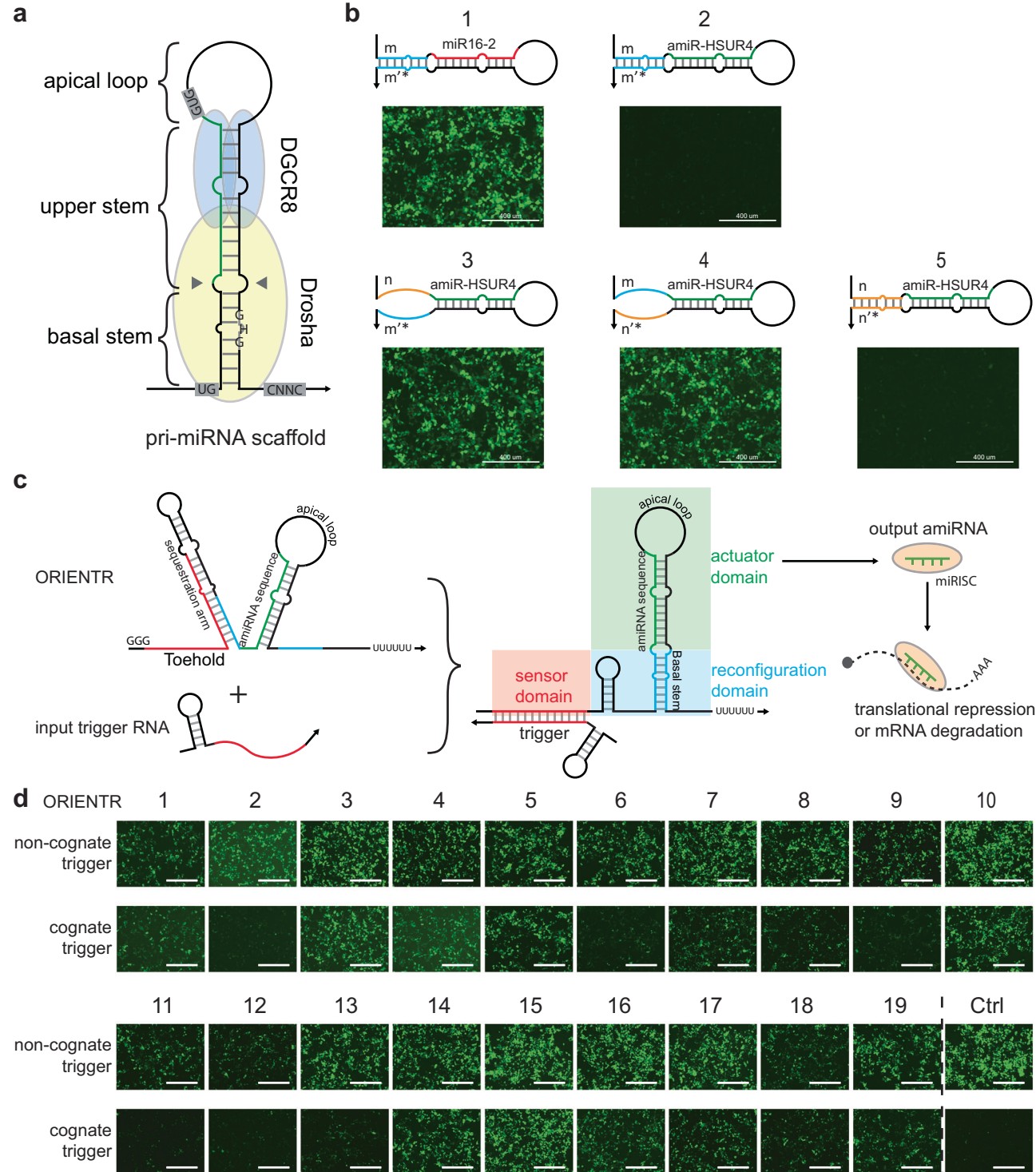

**Fig. 1 | Design and implementation of ORIENTRs based on pri-miRNA substrates. a** Structural features and sequence motifs of pri-miRNA substrates. The green line indicates the sequence that matures into a 5p amiRNA. **b** Investigation of pri-miR-16-2 sequence and structural elements required for RNAi activity: (1) original pri-miR-16-2, (2) the sequence for miR16-2 was swapped with miR-HSUR4, a viral microRNA sequence, to target a GFP reporter, (3) change of the 5′ basal stem m to the scrambled control sequence n to disrupt base-pairing, (4) change of the 3′ basal stem m'* to scrambled control sequence n'* to disrupt base-pairing, and (5) change of the basal stem sequence from m-m'* to n-n'* to reconstruct stem structure. The top diagrams depict the pri-miRNA secondary structures

predicted by RNAfold[55] (see Supplementary Fig. S1e for nucleotide-resolution diagrams). The bottom images show the GFP reporter signal cells transfected with the pri-miRNAs. $n = 3$ biological replicates. **c** ORIENTR design schematic. The sequestration arm prevents the formation of the basal stem (blue). The binding of the trigger to the sensor domain (red) leads to the formation of the basal stem within the reconfiguration domain, which then enables the processing of the actuator domain (green) into a functional amiRNA. **d** Optical microscopy images of 19 different ORIENTR designs with and without expression of their cognate trigger RNAs. ORIENTR activation leads to the production of an amiRNA output targeting the GFP mRNA. $n = 3$ biological replicates. The scale bar represents 400 μm.

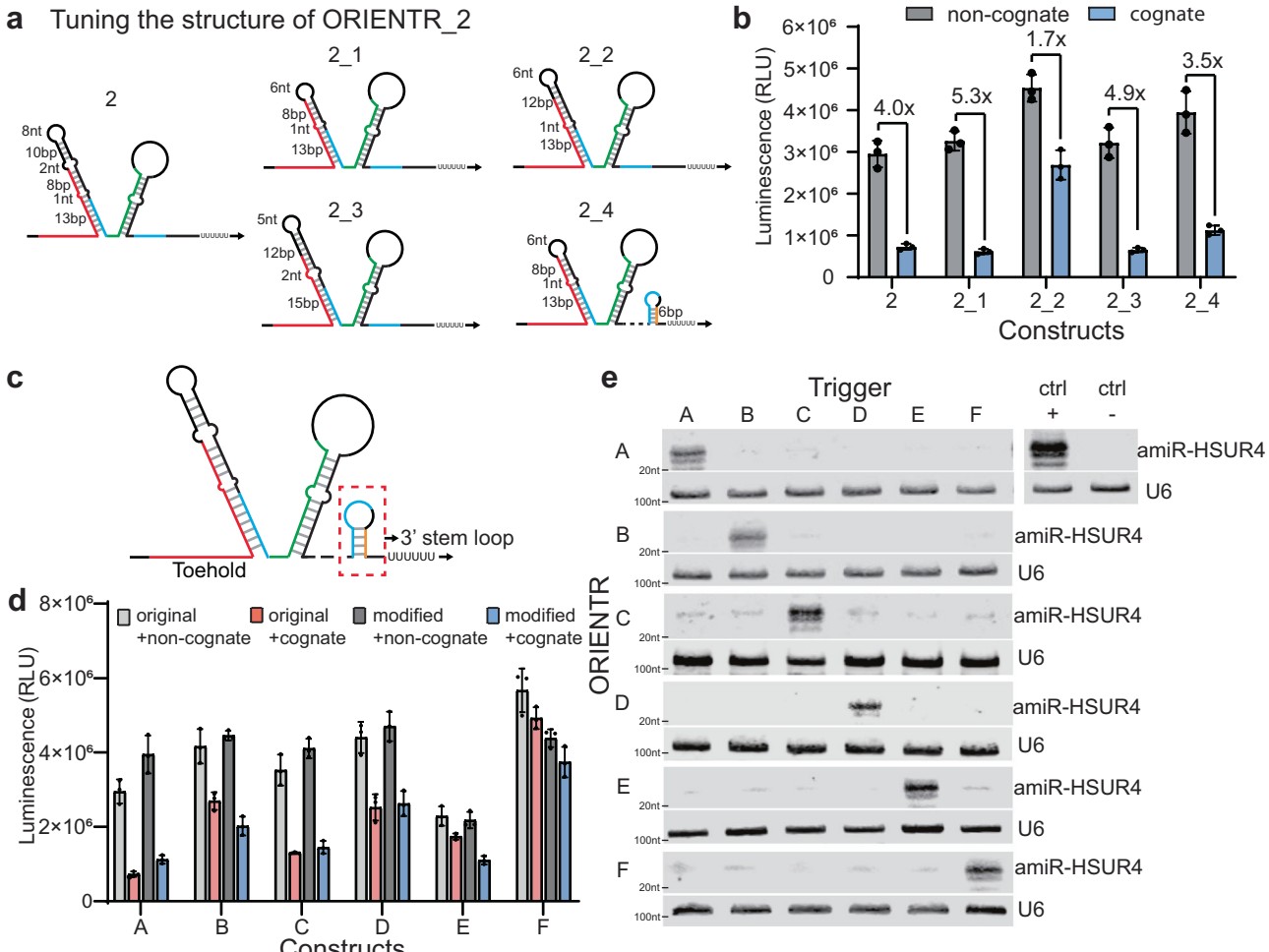

**Fig. 2 | Implementing an orthogonal ORIENTR library with lower leakage via secondary structure tuning. a** RNA secondary structure for ORIENTR_2 and its derivatives. **b** Luminescence signal from cells transfected with ORIENTR_2 and its derivatives with cognate or non-cognate trigger RNAs using a luciferase reporter. $n = 3$ biological replicates, bars represent the mean ± s.d. **c** Diagram of ORIENTR with the 3′ stem-loop (orange), which partially hybridizes to the 3′ domain of the basal stem (blue) to suppress amiRNA biogenesis without the trigger. **d** The performance of six ORIENTRs (ORIENTR_2, 13, 9, 7, 11, 6) before and after the 3′ stem-loop modification from luciferase reporter with cognate or non-cognate triggers. $n = 3$ biological replicates, bars represent the mean ± s.d. **e** Orthogonality test for six optimized ORIENTRs against six triggers with amiRNA northern blot. U6 was probed as the loading control. $n = 3$ biological replicates.

as input through toehold-mediated strand displacement[17] in the sensor domain (Fig. 1c, left). This transition releases the sequestered 11-nt sequence and reconstitutes the basal stem in the reconfiguration domain to activate Drosha recognition, and therefore initiates miRNA biogenesis to generate a mature functional amiRNA as the output (Fig. 1c, right). The binding signal recorded by the sensor domain is transmitted to the actuator domain via structural refolding of the reconfiguration domain, rather than the amiRNA-encoding region, successfully decoupling the input sequence from the output sequence. With de novo computational RNA design by NUPACK[36] (Supplementary Fig. S2), we generated 19 ORIENTR devices with corresponding cognate trigger RNAs (see Supplementary Table S5 for sequences) and constructed them in separate plasmids with transcription initiated by the human U6 promoter. To inhibit RNA degradation, a small protecting hairpin was included at the 5′ end of the trigger RNA[37]. ORIENTR and trigger RNA plasmids, together with a reporter GFP plasmid, were co-transfected into HEK293T cells. For many of the ORIENTRs, cells displayed lower GFP fluorescence in the presence of the cognate trigger compared to the non-cognate trigger (Fig. 1d), and similar results were achieved from a luciferase reporter (Supplementary Fig. S3), indicating a cognate trigger RNA induced RNAi activity.

## Optimizing ORIENTR structure

Although our design strategy proved effective in generating RNAi in response to specific cellular RNAs, we still faced two major challenges: firstly, ORIENTR folding has defects that led some devices to display leaky amiRNA production that repressed the target gene in the absence of cognate trigger RNA (e.g., ORIENTR_12 in Fig. 1d); and secondly, the dynamic range of regulation is limited, suggesting only partial activation of ORIENTRs by the RNA triggers (e.g., ORIENTR_7 in Fig. 1d).

To investigate the effect of ORIENTR structure on performance, we modulated the secondary structure of the sequestration arm by varying the stem length and bulge size as well as the flexibility of 3′ basal stem for ORIENTR_2 (Fig. 2a and Supplementary Table S6). The results show that the conformation of the sequestration arm and the downstream flanking sequence affects signal leakage and the sensitivity of the response to the RNA trigger (Fig. 2b). Thus, it is possible to tune ORIENTR design parameters to achieve minimum background activation or maximum output response to suit specific requirements in certain applications.

We observed that in ORIENTR_2_4, the 6-nt domain that weakly base-paired with the 3′ basal stem decreased leaky expression. Compared with the original construct, the reporter signal in the presence of

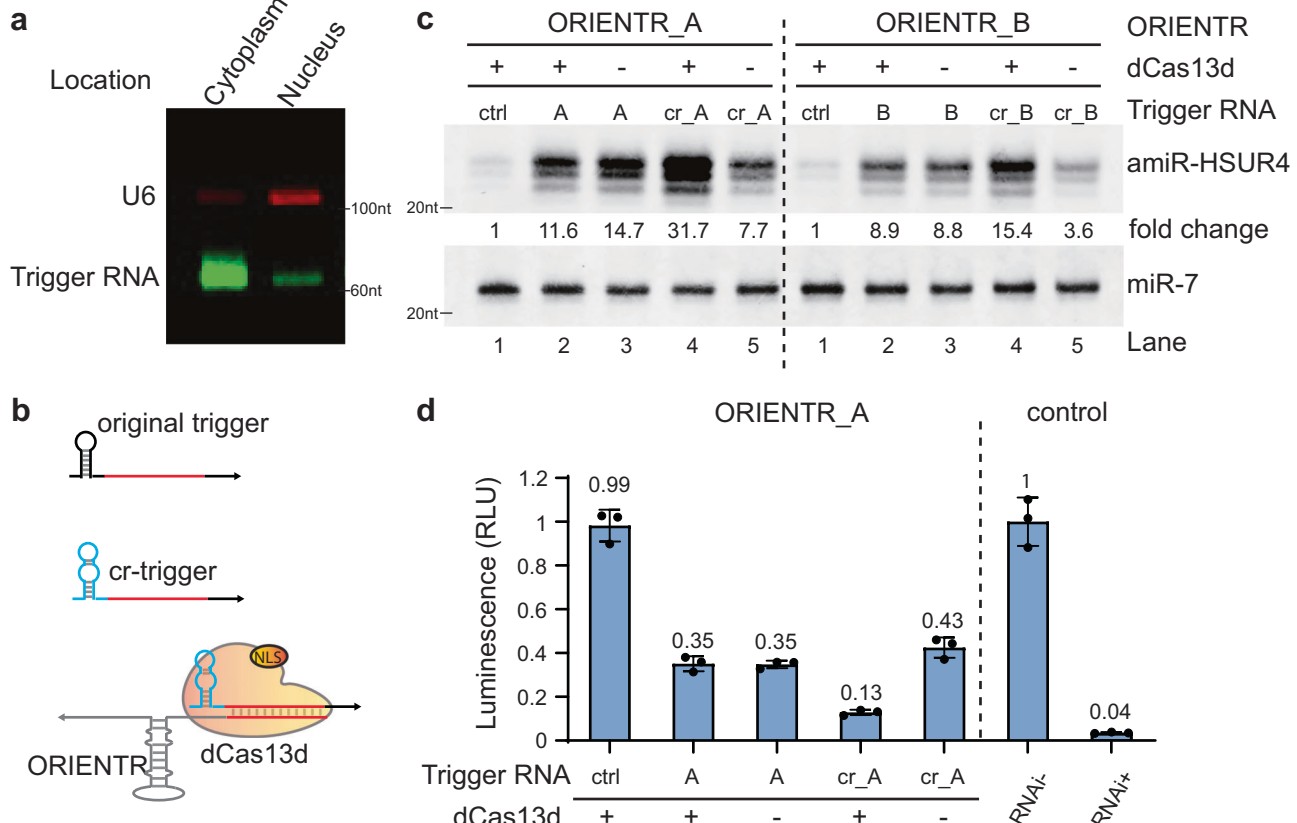

**Fig. 3 | Enhancing ORIENTR dynamic range using dCas13d. a** Profile of RNA triggers cellular localization in the cytoplasm and nucleus. **b** Modified trigger design to change the 5′ stabilizing stem to a crRNA hairpin to form a complex with dCas13d bearing a nuclear localization signal (NLS). **c** Northern blot comparing performance between the original trigger and modified trigger in the presence of dCas13d for two representative ORIENTRs. The numbers below the amiR-HSUR4 bands are the quantitated band intensity after normalization by miR-7 and by the control group. $n = 3$ biological replicates. **d** RNAi assay with luciferase reporter in cells for ORIENTR_A with the original trigger or a cr-trigger with or without dCas13d. $n = 3$ biological replicates, bars represent the mean ± s.d.

ORIENTR_2-4 increased by 34% while preserving responsiveness against its trigger (Fig. 2b). We included this motif (Fig. 2c), which we term the 3′ stem-loop, for another five representative ORIENTR switches (ORIENTR 13, 9, 7, 11, 6). ORIENTR_2-4 and the other five modified ORIENTRs were together renamed as ORIENTRs A through F (Supplementary Table S7). In general, they provided larger changes in reporter expression with and without the trigger compared to designs lacking the 3′ stem-loop (Fig. 2d, Supplementary Fig. S4); however, the effect of this motif varied depending on the specific ORIENTR. To improve the trigger performance, we also studied trigger RNAs containing two small protecting hairpins at both the 5′ and 3′ ends to further inhibit trigger RNA degradation. However, these triggers did not perform as well as those with the hairpin only at the 5′ end (Supplementary Fig. S5 and Supplementary Table 8). Therefore, we continued with the original trigger RNAs.

We then tested the orthogonality of the six ORIENTR devices and their triggers by co-transfecting each ORIENTR with all six triggers individually and using northern blots to quantify the amiRNA production from whole-cell extract (Fig. 2e). Although some ORIENTR devices still showed leaky amiRNA generation, all six devices demonstrated high orthogonality with increased amiRNA production observed only in the presence of the cognate triggers.

**Improving ORIENTR regulation dynamics with dCas13d**
Next, we aimed to increase the regulation dynamics by enhancing ORIENTR interaction with the trigger RNA. Since Microprocessor regulation occurs in the nucleus, we first investigated the localization of ORIENTR and trigger RNA. We noticed that the majority of ORIENTR

(Supplementary Fig. S6) and trigger RNAs (Fig. 3a) were exported to the cytoplasm. We hypothesized that the trigger RNA interaction with ORIENTR was limited by RNA abundance and stability in the nucleus.

To improve the performance of the trigger RNA, we replaced the 5′ protecting hairpin in triggers A and B with the CRISPR RNA (crRNA) scaffold hairpin used by rfxCas13d[38] to form chimeric triggers cr-A and cr-B (see Supplementary Table S9 for sequences) that can be bound by deactivated rfxCas13d (dCas13d) (Fig. 3b). The RNA-dCas13d complex formed between these chimeric CRISPR triggers (cr-triggers) and dCas13d mimics the natural RNA/RBP (RNA-binding protein) interactions in mammalian cells to protect the small trigger RNAs from degradation. Moreover, the use of a nuclear localization signal peptide on dCas13d can help transport the trigger RNA back into the nucleus. From our results, the chimeric triggers alone were less effective at generating amiRNAs than the original triggers (compare lane 5 to lane 3 for ORIENTR_A and B in Fig. 3c), possibly because the crRNA scaffold hairpin is less stable than the original 5′ hairpin at protecting the RNA from degradation. However, in the presence of dCas13d (lane 4 in Fig. 3c), the cr-trigger RNAs significantly promoted amiRNA production, each giving 31.7-fold and 15.4-fold increases in amiRNA levels compared with the non-cognate trigger RNA and significantly higher than the 14.7-fold and 8.8-fold increases observed for the original trigger RNAs without dCas13d present. The RNAi efficiency of ORIENTR_A was also quantitatively measured with a luciferase reporter (Fig. 3d). We found that there is minimal amiRNA biogenesis from ORIENTR alone, with the luciferase activity retained at 99%. The cr-trigger significantly induced RNAi, decreasing luciferase activity to 13%, corresponding to a 7.7-fold reduction in gene expression. The

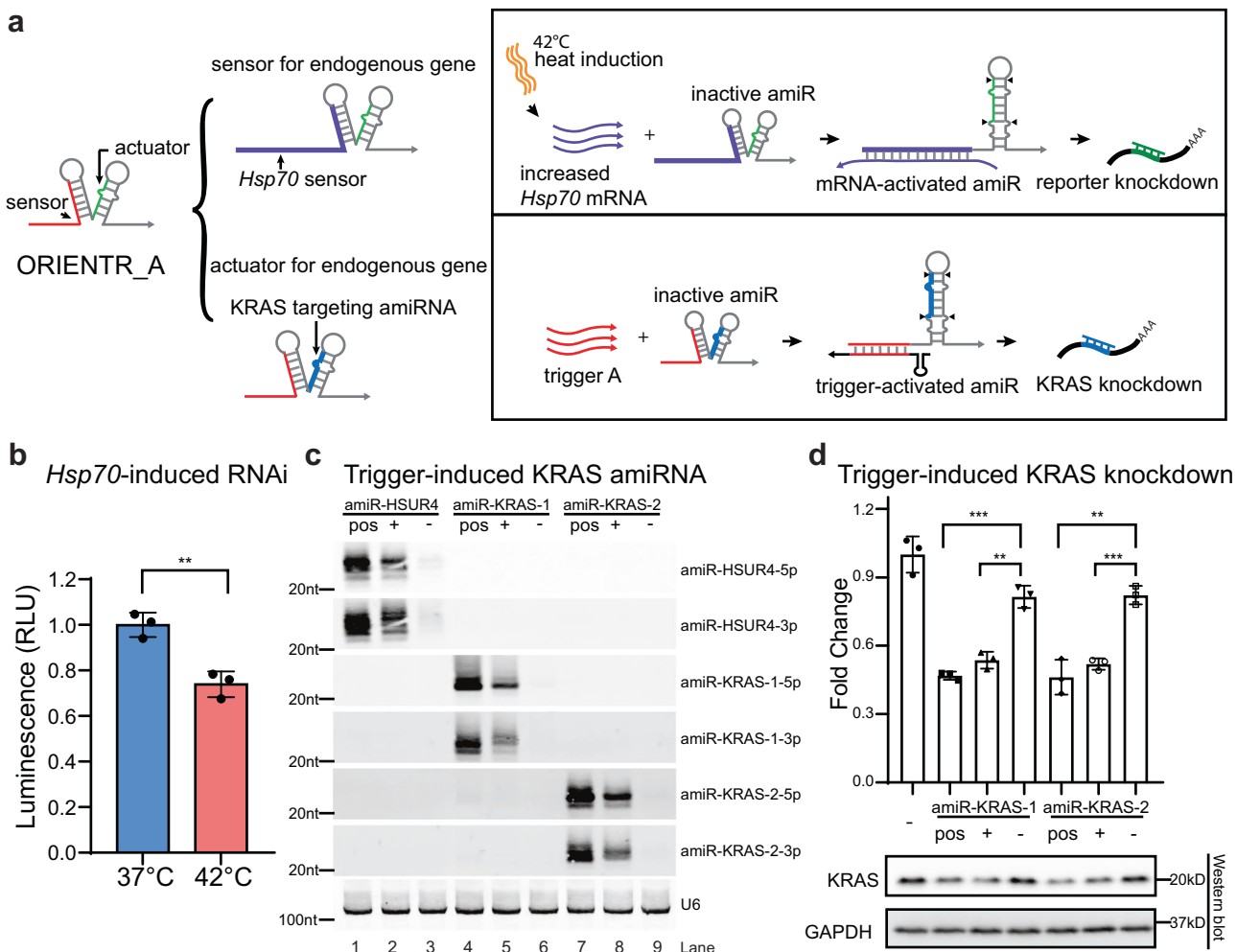

**Fig. 4 | Sensing and regulating endogenous genes using ORIENTRs.**
**a** Schematics for an ORIENTR with an engineered sensor domain against *Hsp70* mRNA (top) or actuator domain against *KRAS* (bottom). **b** Luminescence signal with ORIENTR-*Hsp70* from cells at 37 °C or subjected to heat shock at 42 °C. The signal was normalized by cells transfected with a control plasmid. (∗∗) *P* = 0.0044 < 0.01, unpaired two-tailed *t* test, *n* = 3 biological replicates, bars represent the mean ± s.d. **c** Northern blot analysis of amiR-HUSR4-5p, amiR-HUSR4-3p, amiR-KRAS-1-5p, amiR-KRAS-1-3p, amiR-KRAS-2-5p, amiR-KRAS-2-3p, amiR-16, and U6 in HEK293T cells transfected with positive control of pri-miRNA and ORIENTRs with cognate/non-cognate triggers for amiR-HUSR4, amiR-KRAS-1 and amiR-KRAS-2, respectively. The levels of U6 serve as loading controls. 'Pos' denotes the positive control of pri-miR-HSUR4, pri-miR-KRAS-1, and pri-miR-KRAS-2, respectively. '+' denotes the ORIENTR with its cognate trigger. '−' denotes the ORIENTR with a non-cognate trigger. *n* = 3 biological replicates. **d** RT-qPCR (top) and western blot (bottom) measurement of the levels of KRAS in HEK293T cells transfected with negative RNA control, positive pri-miRNA, and ORIENTR with cognate/non-cognate triggers for amiR-KRAS-1 and amiR-KRAS-2, respectively. GAPDH serves as the loading control. (∗∗) P(amiR-KRAS-1 + /-) = 0.0014 < 0.01, (∗∗∗) P(amiR-KRAS-1 Pos/-) = 0.0003 < 0.001, (∗∗) P(amiR-KRAS-2 Pos/-) = 0.0019 < 0.01, (∗∗∗) P(amiR-KRAS-2 + /-) = 0.0004 < 0.001, unpaired two-tailed *t* test, *n* = 3 biological replicates, bars represent the mean ± s.d.

improvement of cr-trigger/dCas13d compared with the original trigger (35% luciferase activity) implicated that RNA localization and/or stability have a significant impact on ORIENTR performance in mammalian cells. We further investigated trigger RNA subcellular localization and abundance with and without dCas13d by northern blot, and the results showed that trigger RNA level increased by 4- to 5-fold in both the cytoplasm and nucleus in the presence of dCas13d (Supplementary Fig. S7). Our success in incorporating protein components to facilitate programmed RNA interactions suggests that protein-mediated RNA stabilization and localization could be a general strategy for enhancing RNA switch performance in mammalian cells.

## Using ORIENTRs for sensing and knockdown of endogenous mRNAs

Since the ORIENTR design decouples the sequence of its sensor and actuator domains, ORIENTRs can be trained against endogenous transcripts in principle. We first investigated if ORIENTRs can be

applied for sensing endogenous RNA transcripts. The expression of heat shock proteins is tightly regulated in response to stress conditions, and *Hsp70* mRNA abundance can increase substantially after 42 °C heat shock in HeLa cells[39]. To implement an *Hsp70*-responsive ORIENTR, we engineered an ORIENTR with an extended 74-nt toehold to strengthen its interaction with the endogenous transcript and designed the ORIENTR to bind to a region of lower secondary structure in the *Hsp70* mRNA (Fig. 4a top, see Supplementary Table 10 for sequence information). The amiRNA for the *Hsp70*-responsive ORIENTR was used for the knockdown of luciferase. When cells were subjected to heat shock conditions at 42 °C, the reporter signal decreased by 26% relative to levels at 37 °C following normalization with a control ORIENTR (Fig. 4b and see "Methods" for experimental details).

We next applied ORIENTRs for targeting an endogenous gene. *KRAS* is among the most well-known proto-oncogenes, and it is viewed as a promising target for the treatment of multiple cancers[40]. However,

a major difficulty for KRAS therapies is to limit their effects on cancer cells while minimizing their adverse effects on healthy cells[41]. To implement a more targeted KRAS knockdown system, we designed two sets of KRAS-targeting amiRNAs and incorporated them in the scaffold of ORIENTR_A (Fig. 4a bottom and see Supplementary Table 10 for sequence information). Without the cognate trigger, we observed only minimal amiRNA production. In contrast, the presence of trigger RNA significantly increased amiRNA production for both cases (Fig. 4c). Gene knockdown efficiency was evaluated by RT-PCR (Fig. 4d top) and western blot (Fig. 4d bottom), and both showed trigger-induced KRAS suppression. Importantly, we noticed that the biogenesis of passenger amiRNAs from the pri-miR-16-2 scaffold was also regulated by trigger RNA presence, indicating that ORIENTRs help limit off-target effects from passenger amiRNA. Experiments with ORIENTRs featuring actuator amiRNAs for knockdown of *CSN1KD* and two other *KRAS* sites also demonstrated trigger-dependent amiRNA biogenesis (Supplementary Fig. S8 and Supplementary Table S10). Altogether, these results demonstrate that the built-in modularity of ORIENTRs can be applied to sense and regulate intracellular genes of interest.

## Discussion

By systematically examining the sequence and structural requirements for functional pri-miRNAs, we have implemented RNA switches that provide conditional pri-miRNA activity. ORIENTRs employ cis-repressing RNA secondary structures to prevent amiRNA biogenesis, while interactions with cognate RNA triggers stimulate amiRNA biogenesis and activate RNAi against a desired target gene. To do this, ORIENTRs encode within a single RNA molecule a sensor domain, a reconfiguration domain, and an actuator domain that all fold faithfully into a prescribed structure in the complex cellular environment (Fig. 1c). These domains also undergo substantial structural rearrangements upon binding to the RNA trigger to enable recognition by Drosha. The use of sensor and actuator domains that are not correlated in sequence provides ORIENTRs the capacity to direct a tissue-specific signature RNA to a targeted RNA interference response. Using de novo RNA sequence design, we identified six orthogonal ORIENTR/ trigger pairs, demonstrated that signal leakage from ORIENTRs can be tuned through modification to RNA structure, and found that regulation dynamics can be improved by transporting the trigger RNA back to the nucleus with dCas13d (Figs. 2 and 3). By harnessing the independent ORIENTR sensor or actuator domains, we demonstrated that ORIENTRs can detect endogenous mRNA and knockdown endogenous genes of interest, indicating that ORIENTRs could potentially be applied to building feedback loops, or for programming and rewiring cell behavior in response to RNA expression profiles for diagnostic or therapeutic purposes. To our knowledge, our approach is the first demonstration of applying programmable cellular RNAs to regulate RNA interference in living cells and thus represents an important advance toward introducing synthetic regulatory links into mammalian cells and organisms.

We anticipate that ORIENTRs could have several other important applications enabled by the precise control of RNAi activity. By ensuring that RNAi is only activated when desired, they can help avoid over-burdening the endogenous miRNA biogenesis machinery[42] and thus minimize their impact on host cell function. ORIENTRs also provide more cell-specific RNA interference that is regulated by RNA profile to potentially reduce side effects on off-target cells. Researchers have used tissue-specific promoters[43] and AAV serotypes[44,45] to confine RNAi to desired target tissues. However, their availability and specificity are still limited for applications in certain tissues. For instance, they lack specificity in recognizing different types of muscles[46] or nerves[47], or distinguishing healthy and diseased cells. However, ORIENTRs can utilize transcriptomic differences as activation signals that can vary between cell types, along different cell developmental stages, in response to environmental stimuli, or when affected by disease. These transcriptomic differences can provide more accuracy and dynamic regulation to enhance the specificity of interference. Therefore, ORIENTRs could provide an added layer of regulation when used in combination with tissue-specific promoters and AAV delivery and could thus be an enabling tool for next-generation selective and safe AAV-mediated RNAi gene therapy[48–50].

It is important to mention that although our model was based on the pri-miR-16-2 scaffold, in principle, the ORIENTR mechanism could be adapted to any endogenous or synthetic pri-miRNA scaffold whose basal stem region is devoid of sequence constraints. We expect that further optimization of these elements based on more advanced understanding of the pri-miRNA scaffolds[51] and the strand selection process[52] can help to improve amiRNA production dynamic range and quality.

When generating an ORIENTR for a particular RNA trigger molecule, our experience reveals several general principles for ORIENTR design. A stronger sequestration arm helps to decrease leaky amiRNA expression, while a longer toehold in the sensing region can facilitate RNA trigger recognition and interaction. Moreover, the trigger RNA should possess a low secondary structure to facilitate ORIENTR recognition, and ideally, it should be present at a high copy number in the nucleus. Incorporating these principles should maximize the range of triggers and target genes to which ORIENTRs can be applied.

Lastly, the demonstrated ability of de novo designed RNAs for conditional RNAi suggests possible natural mechanisms for RNAi regulation through modulation of pri-miRNA secondary structure. Indeed, miRNA-mediated inhibition of miRNA biogenesis has recently been reported for viral miRNAs[53], raising the intriguing possibility that similar natural regulatory interactions could occur in cellular small RNAs.

## Methods

### ORIENTR computational design

Designs for the ORIENTR and triggers were generated using the NUPACK complex design package[36], which enabled simultaneous structural optimization of the ORIENTR, trigger, and trigger–ORIENTR complex. The prevented sequences were AAAA, CCCC, GGGG, UUUUU, KKKKKK, MMMMMMM, RRRRRRR, SSSSSSS, WWWWWWW, and YYYYYY. The test tube capability or pseudoknot setting was not used in the designs. See Supplementary Note 1 document for the NUPACK design script used for generating the ORIENTRs.

### Plasmid construction

ORIENTR and trigger plasmids were constructed by site-directed mutagenesis (NEB Catalog No. E0054S). A Bluescript SK plasmid with a human U6 promoter was linearized by PCR with primers containing the insert fragment and then sealed back by blunt end joining. The luciferase reporter plasmid was generated from the pmirGLO Dual-Luciferase miRNA Target Expression Vector (Promega catalog No. E1330) by inserting two miR-HSUR4 target sites between SacI and NheI at the 3′ end of the firefly luciferase gene. For generation of the GFP reporter plasmid, two miR-HSUR4 target sites were inserted into pTYF-U6-stuffer-PGK-EGFP via KpnI sites. All oligonucleotides were purchased from Integrated DNA Technologies (IDT).

### Cell culture and transfection

HEK293T (ATCC, CRL-3216) and HeLa (ATCC, CRM-CCL-2) cells were cultured in DMEM with 1% penicillin and streptomycin, and 10% fetal bovine serum at 37 °C with 5% $CO_2$. For plasmid transfection experiments, $2.5 \times 10^5$ cells/well of HEK293T cells were seeded in 12-well plates. For the GFP reporter assay, 1.4 μg of plasmids (0.4 μg ORIENTR plasmid, 0.8 μg trigger plasmid, 0.2 μg GFP reporter plasmid) were transfected into one well with 2.5 μL Lipofectamine 3000 (Invitrogen) according to the manufacturer's instructions. For dCas13d

experiments, 1.4 μg of plasmids (0.4 μg ORIENTR plasmid, 0.8 μg trigger plasmid, 0.2 μg dCas13d plasmid) were transfected into one well.

## Northern blot
Northern blots using near-infrared probes were performed as described previously[54]. Total RNA was extracted by Trizol reagent (Invitrogen, 15596026) 48 h after transfection, and separated on 15% Urea PAGE, then transferred to Hybond-NX membrane (GE Healthcare, RPN303T). Probes are listed in Supplementary Table 11.

## Luciferase assay
50 ng of ORIENTR plasmid, 100 ng of trigger plasmid, and 25 ng of luciferase reporter plasmid were transfected into HEK293T cells. The luciferase assay was performed with the Dual-Glo® Luciferase Assay System from Promega (Catalog No. E2920) according to the manufacturer's protocol. The Luciferase signal was read with a plate reader (BMG Labtech, CLARIOstar), and firefly luciferase was normalized by Renilla luciferase to calculate the signal in terms of relative luminescence units (RLU). For the dCas13d experiment in Fig. 3d, the firefly luciferase signal was then normalized by a signal from the RNAi(-) control.

## Nuclear and cytoplasmic cell fractions
Cells were gently resuspended in Sucrose Buffer I (0.32 M/10.94% sucrose, 3 mM CaCl$_2$, 2 mM Mg acetate, 0.1 mM EDTA, 10 mM Tris-HCl pH 8.0, 0.3 % NP-40 supplemented with 1x Protease Inhibitor (PI), 1 mM NaF, and 1 mM Na$_3$VO$_4$). Subsequently, cells were centrifuged at $600 \times g$ at 4 °C for 3 min, and the supernatant was kept as cytoplasmic extract. The pellet containing nuclei was gently washed three times with Sucrose Buffer I without NP-40 by pipetting and centrifuging at $300 \times g$ at 4 °C for 3 min. Total RNA from nuclear and cytoplasmic fractions were extracted by Trizol reagent (Invitrogen, 15596026).

## Heat shock assay
HeLa cells from ATCC (CCL-2) were maintained in DMEM supplemented with 10% fetal bovine serum and 1% penicillin and streptomycin at 37 °C with 5% CO$_2$. For heat shock sensing, cells were plated onto a 96-well plate one day before transfection. When cells reached 80% confluence, transfection was performed using the TransIT-MONSTER® Transfection Kit (MIR 2900 from Mirus Bio) according to the manufacturer's protocol. For each well, 120 ng of plasmid (80 ng ORIENTR plasmid and 40 ng luciferase reporter plasmid) was transfected. ORIENTR_A was used as a control plasmid. 24 h post-transfection, half of the cells were moved to 42 °C (5% CO$_2$). After 24 h of heat shock, cells were harvested and subjected to luciferase measurement. The relative light units (RLU) at 42 °C were then calculated as below: fluc signal was first normalized by rluc signal, and then normalized by ORIENTR_A, followed by normalization by value at 37 °C. fluc37 and rluc37 are the luminescence intensities from firefly luciferase and Renilla luciferase, respectively, at 37 °C. fluc42 and rluc42 are the luminescence intensities from firefly luciferase and Renilla luciferase, respectively, at 42 °C.

$$ RLU = \frac{\dfrac{fluc42\left(ORIENTR_{Hsp70}\right)/rluc42\left(ORIENTR_{Hsp70}\right)}{fluc42\left(ORIENTR_A\right)/rluc42\left(ORIENTR_A\right)}}{\dfrac{fluc37\left(ORIENTR_{Hsp70}\right)/rluc37\left(ORIENTR_{Hsp70}\right)}{fluc37\left(ORIENTR_A\right)/rluc37\left(ORIENTR_A\right)}} $$

## Western blot
To detect the protein level of KRAS, HEK293T cells transfected with negative RNA control, positive pri-miRNA and ORIENTR with cognate/non-cognate trigger for amiR-KRAS-1 and amiR-KRAS-2 were lysed by NP-40 lysis buffer (50 mM Tris-HCl pH 7.5, 1% NP-40, 10% glycerol,

150 mM NaCl, 5 mM EDTA, and 0.5 mM PMSF) for 30 min at 4 °C and supernatant was collected by centrifugation at $21,000 \times g$ for 10 min at 4 °C. The supernatant was subjected to a western blot. Antibodies for KRAS (Thermo Fisher Scientific, 703345) and GAPDH (Sigma, G9545) were used for Western blots with 1:2500 dilutions.

## Statistics & reproducibility
No statistical method was used to predetermine the sample size. No data were excluded from the analyses; randomization and blinding are not relevant to this study.

## Reporting summary
Further information on research design is available in the Nature Portfolio Reporting Summary linked to this article.

## Data availability
The main data supporting the results of this study are available within the paper and the Supplementary Information. The datasets generated during and/or analyzed during the current study are available in the Source Data file provided in this paper. Source data are provided in this paper.

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

## Acknowledgements
This work was supported by an NIH Director's New Innovator Award (1DP2GM126892), an NIH R01 award (1R01EB031893), and Boston University startup funds to A.A.G., and an NIH grant (R35GM128753) to M.X. The content is solely the responsibility of the authors and does not necessarily represent the official views of the National Institutes of Health.

## Author contributions
Y.Z., P.S., M.X., and A.A.G. conceived the project and interpreted the data. Y.Z., P.S., J.L., and Y.L. performed the experiments. J.L and Y.L. developed the *Hsp70*-responsive ORIENTRs and tested ORIENTR triggers with 5′ and 3′ hairpin structures. Y.Z., P.S. performed the rest of the experiments and wrote the paper. M.X. and A.A.G. edited the paper and supervised the research.

## Competing interests
A.A.G. is a co-founder of En Carta Diagnostics, Inc. Y.Z., P.S., M.X., and A.A.G. have filed an international patent application (PCT/US23/85466) and a U.S. patent application (18/393,408) pertaining to ORIENTRs. The remaining authors declare no competing interests.
