## [Peer Review File · Nature Communications]

REVIEWER COMMENTS

Reviewer #1 (Remarks to the Author):

Zhou et al.

Conditional RNA interference in mammalian cells via RNA transactivation

In this short manuscript, the authors have engineered a pri-miRNA scaffold to generate a switchable small RNA expression tool, which they refer to as ORIENTR. They used a pri-miR-16 scaffold and first replaced the miRNA sequence by a viral miRNA sequence in order to distinguish it from endogenous miR-16. The authors then extended the 5' and also the 3' ends by structures that unfold the basal stem of the pri-miRNA hairpin, which inhibits microprocessor cleavage and thus this molecule does not generate a small RNA and remains silent. However, when adding a trigger RNA, the complementary basal stem is released, folds and can be processed. A short stem on the 3' end decreases leakage without the trigger RNA. To further improve the system, the authors engineered their ORIENTR platform by adding complementary guide RNA sequence that can be used to recruit dCas13d through a cr-trigger. This stabilizes the systems and increases small RNA production and the silencing output.

This is an interesting manuscript and innovative approach. It is nevertheless difficult to imagine that such a complex tool would be used for gene therapy in humans since it would require the expression and delivery of at least three individual components. However, it could be an interesting tool for synthetic biology purposes and thus the study has its merits. Listed below, there are a number of issues that are unclear or require further control.

1. Figure 1b: these constructs are probably also expressed from plasmids? This is not fully clear from the text of the results section and the legend. Are only the trigger RNAs expressed from plasmids and the pri-miRNA scaffold as RNA? This needs to be explained better.

2. Figure 1b: the wording "scrambled sequences" somehow intend that a larger pool or library is used for some kind of screening approach. In fact, there are only three constructs generated. This should be explained and rationalized better. Furthermore, the main Figure uses m'* and n'* and the suppl. Figure n* and m*? This is somewhat confusing.

3. The authors generate a single stranded trigger RNA. To stabilize it, they included a stem-loop structure. It is nevertheless surprising that such an RNA is stable in a cell. Half-lives of these RNAs might be very low. Why not protecting it on both ends?

4. The authors hypothesize that the trigger RNA becomes exported from the nucleus, which is rather speculative. It could also be more degraded by nuclear ribonucleases. Adding the dCas13d systems is certainly a clever approach. However, does it increase nuclear localization? This could easily be tested and should be included. Why do the Northern blot signals look different in nuclear and cytoplasmic fractions?

Reviewer #2 (Remarks to the Author):

In the manuscript by Zhou Yu et al., the authors developed a method for conditional induction of RNA interference in human cells using short RNA as a trigger of miRNA biogenesis. The system is composed of modified pri-miR16-2 scaffold in which the 5' half of the basal stem forms hairpin structure to suppress miRNA biogenesis without the trigger. Addition of the trigger RNA opens up the structure and miRNA biogenesis can be initiated by Microprocessor complex. To increase nuclear localization of a trigger and its stability the authors used dCas13d and chimeric trigger RNA containing CRISPR RNA scaffold hairpin. At first glance, the idea seems interesting, but in my opinion it has several weaknesses. First, the system composed of few components (modified RNAi construct, cr trigger-RNA and dCas13d) is quite complex and to be effective it has to be delivered to the same cell in an optimal ratio. Secondly, vector-based trigger RNA cannot be turned off. A more interesting system and more needed from the point of view of, for example, AAV-based RNAi therapy, would be to turn off RNAi (e.g., in case of side effects in patients or too much silencing). RNAi activity in the relevant tissues is currently ensured with the appropriate AAV serotypes and tissue-specific promoters. In this aspect, too, I do not see the advantage of the proposed solution. In terms of experimentation, the work also has many shortcomings: (i) there is no demonstration of the effect on the endogenous locus - only reporter systems were used, (ii) no demonstration of ORIENTR efficiency and precision using various inserts/targets, (iii) since Drosha processing is important feature in the overall activity of the system, the analysis of a quality and quantity of the products should be performed (small RNAseq), (iv) the issue of off-target/non-specific effects is not addressed in this study.

Major

1. It has previously been shown that pri-miRNAs are very sensitive to modifications in sequence and structure, such as when 21 bp of miRNA is exchanged for target-specific siRNA. Even minimal changes can affect the quality and quantity of pri-miRNA post-processing products. Therefore, it is difficult to agree with the statement that since „ORIENTRs completely decouple the trigger RNA sequence from output miRNA sequence .. an arbitrary RNA input can be used to silence any desired mRNA”. The exchange of approximately 40 nt sequences may have a significant impact on the structure of the entire construct and its processing. Without demonstrating this experimentally through the use of different inserts/targets, the authors' statement is unjustified.

2. The ORIENTR system is based on pri-miR-16-2 (although it is not commonly used as a amiRNA scaffold). The authors justify that: „We identified the basal stem of the pri-miRNA, which adopts a conserved structure with relatively flexible sequence requirements”. Could they write something more about the process of selection? Are the pri-miR-30 or pri-miR-155 scaffolds (commonly used in RNAi technology) not suitable for such modifications? Can a similar system be proposed for other scaffolds (if only based on structure prediction)?

3. Although the proposed system is very interesting, its characterization has been limited to analyzing its effectiveness in simple reporter systems on a single exogenous target. The authors achieve a high level of expression of all components of the system and high silencing efficiency. However, it is important to demonstrate ORIENTR-based regulation of RNAi using the endogenous locus as a target (over time).

4. Northern blots results are confusing – miRNAs are present in the nucleus and ORIENTR pri-miRNA precursors in the cytoplasm (FIG. S5). A very strong signal may indicate saturation of the miRNA biogenesis pathway.

5. Since the aim of the study was to control RNAi by inducing pri-miRNA processing, a better analysis of processing (kinetics, quality and quantity of products from both arms of the hairpin, since the guide/passenger ratio is important for off-target activity) would be necessary.

6. The proposed system consists of several elements, each of which may cause side effects. In the minimal version of the experiment at least off target analysis for trigger RNA should be performed.

7. Introduction: Since there are many names for RNAi technology tools in the literature, describing them precisely can help avoid confusion. So, to be precise, the ORIENTR system consists of a conditional pri-miRNA scaffold containing an artificial miRNA insert that can be designed to target any gene of interest (miRNAs, endogenous regulators of gene expression, have a precisely defined nucleotide sequence). By "miRNA delivery" one can mean the delivery of miRNA mimetics/mimics to replenish the level of missing miRNAs in cells (the opposite effect is provided by antimiRs).

8. It should be mentioned in the introduction that "spatiotemporal control of RNAi" can also be controlled by the use of cell type-specific promoters or the viral vector serotype used, etc.,

9. The authors discuss the various possibilities of using the ORIENTR system in very general terms. In my opinion, the presented examples are not convincing and, most importantly, they are not supported by the results. It would be very valuable to demonstrate the usefulness of the ORIENTR system on a real example with a visible biological effect.

Minor

1. Page 2 - I guess the authors meant “systemic” instead of “systematic delivery”
2. Page 4 – „a viral miRNA sequence”, more specifically, it is an artificial miRNA targeting viral HSUR4
3. Page 5, line 5, line 6: „pri-miRNA-16-2” instead of „pri-miRNA” (these observations are not of a general nature but concern a specific case - pri-miR-16-2)
4. Page 5 – „the scaffold lost its capacity to produce miRNA”- this is only a suspicion as the authors did not analyze processing efficiency, guide/passenger strand ratio, etc.
5. The "Materials and methods" section does not contain important information, e.g. regarding the Northern Blot method; Luciferase assay – no information on the concentration of plasmids used.
6. Fig.2b- What is the percentage improvement in "leakage" of the proposed system compared to the original ORIENTR_2?
7. Fig. 2d – the graph does not show a significant improvement in "leakage" after adding the mentioned motif compared to the original molecule
8. Based on the results obtained, can a general principle for creating working pairs of trigger RNA and pri-miRNA be created?
9. error in the reference - in the conclusions, the last paragraph is 3738 instead of - 37, 38; similar, in the first paragraph of the first chapter, 2825, instead of 25, 28.
10. Fig. S3 – no red arrows mentioned

Reviewer #3 (Remarks to the Author):

The authors report a conditional RNAi system that they call Orthogonal RNA Interference induced by Trigger RNA (ORIENTR), that provides a 14-fold increases in miRNA biogenesis upon activation in orthogonal libraries. They show that coupling ORIENTR triggers with Cas13d enhances dynamic range up to 31-fold. They report ORIENTRs that can direct a tissue-specific signature RNA to a targeted RNA interference.

response. They claim that this is the first demonstration of applying programmable cellular RNAs to regulate RNA interference in living cells. As far as I know, this is correct.

They claim that this represents an advance toward introducing synthetic regulatory links into mammalian cells and organisms. I agree.

Overall this paper makes a major step forward and is suitable for publishing in Nature Comm.

Reviewer #1 (Remarks to the Author):

Zhou et al.

Conditional RNA interference in mammalian cells via RNA transactivation

In this short manuscript, the authors have engineered a pri-miRNA scaffold to generate a switchable small RNA expression tool, which they refer to as ORIENTR. They used a pri-miR-16 scaffold and first replaced the miRNA sequence by a viral miRNA sequence in order to distinguish it from endogenous miR-16. The authors then extended the 5' and also the 3' ends by structures that unfold the basal stem of the pri-miRNA hairpin, which inhibits microprocessor cleavage and thus this molecule does not generate a small RNA and remains silent. However, when adding a trigger RNA, the complementary basal stem is released, folds and can be processed. A short stem on the 3' end decreases leakage without the trigger RNA. To further improve the system, the authors engineered their ORIENTR platform by adding complementary guide RNA sequence that can be used to recruit dCas13d through a cr-trigger. This stabilizes the systems and increases small RNA production and the silencing output.

This is an interesting manuscript and innovative approach. It is nevertheless difficult to imagine that such a complex tool would be used for gene therapy in humans since it would require the expression and delivery of at least three individual components. However, it could be an interesting tool for synthetic biology purposes and thus the study has its merits. Listed below, there are a number of issues that are unclear or require further control.

1. Figure 1b: these constructs are probably also expressed from plasmids? This is not fully clear from the text of the results section and the legend. Are only the trigger RNAs expressed from plasmids and the pri-miRNA scaffold as RNA? This needs to be explained better.

Both trigger and pri-miRNA are expressed from plasmids. To make this point clearer, we have updated several sections:

In the second paragraph under “Investigation of functional pri-miRNA requirement”, we changed the line from “the pri-miRNA was repurposed to silence a green fluorescent protein (GFP) reporter” to “the pri-miRNA was transcribed from a pol III promoter (U6 promoter) and repurposed to silence a green fluorescent protein (GFP) reporter”.

Under “ORIENTR design and performance”, the original text specified “we generated 19 ORIENTR devices with corresponding cognate trigger RNAs (see Supplementary Table S5 for sequences) and constructed them in separate plasmids with transcription initiated by the human U6 promoter”.

2. Figure 1b: the wording “scrambled sequences” somehow intend that a larger pool or library is used for some kind of screening approach. In fact, there are only three constructs generated. This should be explained and rationalized better. Furthermore, the main Figure uses m^{*} and n^{*} and the suppl. Figure n^{*} and m^{*}? This is somewhat confusing.

In the main text, the term “scrambled sequence” has been changed to “scrambled control sequence”. The sequences were generated by NUPACK by randomization to create new basal stems that meet the same structural requirements but have different sequences.

n^* and m^* in the main figure were corrected to n'^* and m'^* to be consistent. They represent imperfect base pairing with n and m .

3. The authors generate a single stranded trigger RNA. To stabilize it, they included a stem-loop structure. It is nevertheless surprising that such an RNA is stable in a cell. Half-lives of these RNAs might be very low. Why not protecting it on both ends?

To answer this question, we took the six triggers of ORIENTRs A-F and added a 3' hairpin structure to generate a transcript protected by stem loops at both ends (see Fig. S5). Unfortunately, we found that the additional 3' hairpin led to decreased miRNA production. As a result, we have decided to continue using trigger RNAs only with 5' hairpin protective groups.

4. The authors hypothesize that the trigger RNA becomes exported from the nucleus, which is rather speculative. It could also be more degraded by nuclear ribonucleases. Adding the dCas13d systems is certainly a clever approach. However, does it increase nuclear localization? This could easily be tested and should be included. Why do the Northern blot signals look different in nuclear and cytoplasmic fractions?

As suggested, we tested RNA abundance and localization and these new results are now shown in Fig. S7. dCas13d protected the RNA trigger from degradation both in cytoplasm and nucleus; however, it did not change the RNA abundance ratio between cytoplasm and nucleus. In the revised manuscript, we added this observation in paragraph two under "Improving ORIENTR regulation dynamics with dCas13d": "We further investigated trigger RNA subcellular localization and abundance with and without dCas13d by northern blot (Fig. S7), and the results showed that trigger RNA level increased by 4- to 5-fold in both the cytoplasm and nucleus in the presence of dCas13d (Fig. S7)."

The miRNA northern blot difference between nuclear and cytoplasmic fractions (Fig. S6) could be due to the cytoplasmic-specific nuclease Dicer. The RNA smear could be an intermediate product with single cut or with non-homogenous ends after pre-miRNA been processed by Dicer.

Reviewer #2 (Remarks to the Author):

In the manuscript by Zhou Yu et al., the authors developed a method for conditional induction of RNA interference in human cells using short RNA as a trigger of miRNA biogenesis. The system is composed of modified pri-miR16-2 scaffold in which the 5' half of the basal stem forms hairpin structure to suppress miRNA biogenesis without the trigger. Addition of the trigger RNA opens up the structure and miRNA biogenesis can be initiated by Microprocessor complex. To increase nuclear localization of a trigger and its stability the authors used dCas13d and chimeric trigger RNA containing CRISPR RNA scaffold hairpin. At first glance, the idea seems interesting, but in my opinion it has several weaknesses. First, the system composed of few components (modified RNAi construct, cr trigger-RNA and dCas13d) is quite complex and to be effective it has to be delivered to the same cell in an optimal ratio. Secondly, vector-based trigger RNA cannot be turned off. A more interesting system and more needed from the point of view of, for example, AAV-based RNAi therapy, would be to turn off RNAi (e.g., in case of side effects in patients or too much silencing). RNAi activity in the relevant tissues is currently ensured with the appropriate AAV serotypes and tissue-specific promoters. In this aspect, too, I do not see the advantage of the proposed solution. In terms of experimentation, the work also has many shortcomings: (i) there is no demonstration of the effect on the endogenous locus - only reporter

systems were used, (ii) no demonstration of ORIENTR efficiency and precision using various inserts/targets, (iii) since Drosha processing is important feature in the overall activity of the system, the analysis of a quality and quantity of the products should be performed (small RNAseq), (iv) the issue of off-target/non-specific effects is not addressed in this study.

Major

1. It has previously been shown that pri-miRNAs are very sensitive to modifications in sequence and structure, such as when 21 bp of miRNA is exchanged for target-specific siRNA. Even minimal changes can affect the quality and quantity of pri-miRNA post-processing products. Therefore, it is difficult to agree with the statement that since „ORIENTRs completely decouple the trigger RNA sequence from output miRNA sequence .. an arbitrary RNA input can be used to silence any desired mRNA”. The exchange of approximately 40 nt sequences may have a significant impact on the structure of the entire construct and its processing. Without demonstrating this experimentally through the use of different inserts/targets, the authors' statement is unjustified.

To address this issue, we have performed additional experiments demonstrating that the pri-miRNA 16-2 scaffold can be used to generate multiple amiRNAs. As shown in Fig. 4 of the revised manuscript, pri-miRNA 16-2 can be used to produce two amiRNAs to target *KRAS*. We have also used it for other target genes or locations that include an amiRNA for *CSNK1D* gene and two additional amiRNAs for *KRAS* gene (see Fig. S8). Thus far, we have not encountered any problems using this scaffold for successful amiRNA production.

2. The ORIENTR system is based on pri-miR-16-2 (although it is not commonly used as a amiRNA scaffold). The authors justify that: „We identified the basal stem of the pri-miRNA, which adopts a conserved structure with relatively flexible sequence requirements”. Could they write something more about the process of selection? Are the pri-miR-30 or pri-miR-155 scaffolds (commonly used in RNAi technology) not suitable for such modifications? Can a similar system be proposed for other scaffolds (if only based on structure prediction)?

We chose pri-miR 16-2 as the scaffold because of its knockdown potency (as shown in Fig. S1b and S1c) and also the clear understanding of its interaction with Microprocessor based on crystal structure data¹. In principle, the ORIENTR system could be applied to other functional scaffolds. However, this extension would require additional investigation since different scaffolds may have different levels of dependence on the basal stem and flanking motifs, such as the 5' UG motif and GHG motif within basal stem. The dependence of these motifs will affect the flexibility of the system for accommodating different trigger RNAs.

In revised version of Fig. S1b, we compared pri-miRNA 16-2 scaffold with the pri-miRNA 33 scaffold² (with luciferase reporter) and in Fig. S1c we compared it with pri-miRNA 30a or pri-miRNA 30E³ (with GFP reporter). pri-miR-16-2 showed better knockdown efficiency compared with all three other scaffolds in our experimental conditions.

As pointed out by this reviewer in comment #1, the pri-miR-30E scaffold did not give apparent GFP knockdown, meaning this scaffold is sensitive to sequence exchange. Accordingly, we continued using the pri-miR-16-2 scaffold in later designs.

As discussed above, we added supporting data and references in paragraph two under “Investigation of functional pri-miRNA requirements”: “We further confirmed that the knockdown

efficiency from pri-miR-16-2 scaffold is independent of promoter and is comparatively more potent than other scaffolds^{2,3} using a luciferase reporter or GFP reporter as the target gene (Fig. S1b and Fig. S1c, see Supplementary Table S2 for sequences).”

We also added a statement in the Conclusion: “It is important to mention that although our model was based on pri-miR-16-2 scaffold, in principle it could be adapted to any endogenous or synthetic pri-miRNA scaffold whose basal stem region is devoid of sequence constraints.”

3. Although the proposed system is very interesting, its characterization has been limited to analyzing its effectiveness in simple reporter systems on a single exogenous target. The authors achieve a high level of expression of all components of the system and high silencing efficiency. However, it is important to demonstrate ORIENTR-based regulation of RNAi using the endogenous locus as a target (over time).

In the revised manuscript, we now demonstrate that ORIENTR can be applied to regulate RNAi directed toward endogenous genes (Fig. 4). The results of these new experiments are described in paragraphs under “Using ORIENTR for sensing and knockdown of endogenous RNAs”. By changing the sensor region to target *Hsp70* mRNA, we demonstrated that endogenous RNA transcripts can be used as trigger for ORIENTR actuation. By changing the actuator region to KRAS-targeting amiRNA, ORIENTR has been demonstrated to be capable of knockdown of endogenous genes.

Since the focus of this study is to demonstrate the proof-of-concept from a synthetic biology standpoint, we feel establishing a stably expressed system to repress the endogenous locus over time is beyond the scope this work.

4. Northern blots results are confusing – miRNAs are present in the nucleus and ORIENTR pri-miRNA precursors in the cytoplasm (FIG. S5). A very strong signal may indicate saturation of the miRNA biogenesis pathway.

Yes, amiRNAs were found in the nuclear fraction, which could be caused by either incomplete cell fractionation (U6 RNA was also found in both fractions) or nuclear Ago shuttling amiRNA back to the nucleus^{4,5}. From Fig. S6 (originally Fig. S5), we can see that the endogenous miR-16 can also be observed in both fractions and with a large proportion in cytoplasm.

ORIENTR pri-miRNA was seen both in the cytoplasm and the nucleus. This observation was expected since the ORIENTR is different from “always-on” pri-miRNA in that it is originally inactive and must interact with trigger RNA in nucleus to reform into an active primary RNA that can be processed by Microprocessor. ORIENTRs that do not encounter a trigger remain inactive and are transported into cytoplasm. Since large fraction of ORIENTRs are inactive, it is not likely that it saturates the miRNA biogenesis pathway.

In the Conclusion, we include the statement that: “By ensuring that RNAi is only activated when desired, they can help avoid over-burdening the endogenous miRNA biogenesis machinery⁶ and thus minimize their impact on host cell function.”

5. Since the aim of the study was to control RNAi by inducing pri-miRNA processing, a better analysis of processing (kinetics, quality and quantity of products from both arms of the hairpin, since the guide/passenger ratio is important for off-target activity) would be necessary.

From Fig. 4c, we can see that both arms of the pri-miRNA can produce amiRNAs, and production of both amiRNAs is regulated by interactions with the trigger. The quality and extent of off-target activity of amiRNAs can vary depending on the specific amiRNA that is produced. For example the reporter amiRNA-HSUR4 has three different sizes while amiRNA-KRAS only have two sizes with difference in major size.

The guide/passenger ratio is also greatly affected by the thermostability of the RNA duplex ends and/or by nucleotide compositions which affect RNA interaction with Argonaute and therefore strand selection⁷. Several different models are either complementary or sometimes contradictory and currently there is no clear guidance in how to predict or optimize guide/passenger ratio. Therefore, we believe that optimization of guide/passenger ratio is beyond the scope of this study. However, these effects suggest that deliberate amiRNA sequence choice should be considered when they are applied to desired gene targets to ensure optimal performance. Based on the above discussion, we have added to the Conclusion the following: "We expect that further optimizations of pri-miRNA scaffolds based on more advanced understanding of the pri-miRNA scaffolds⁸ and the strand selection process⁷ can help to improve the amiRNA production dynamic range and quality."

6. The proposed system consists of several elements, each of which may cause side effects. In the minimal version of the experiment at least off target analysis for trigger RNA should be performed.

In the revised manuscript, the trigger RNA was replaced by an endogenous mRNA, and it achieves mRNA-activated target gene knockdown. These results demonstrate that ORIENTRs can respond to trigger RNAs that are naturally present in the cell, alleviating concerns related to off-target trigger effects.

Off-target effects vary by which RNA is used as the trigger and which amiRNA is produced. It is important to choose an RNA trigger that is specific to the target cells, and to choose optimized amiRNA sequences to knockdown specific genes. Depending on particular applications the system to be applied to, the mRNA trigger sensing region and amiRNA production region are both modular and can be tuned to suit experimental needs.

7. Introduction: Since there are many names for RNAi technology tools in the literature, describing them precisely can help avoid confusion. So, to be precise, the ORIENTR system consists of a conditional pri-miRNA scaffold containing an artificial miRNA insert that can be designed to target any gene of interest (miRNAs, endogenous regulators of gene expression, have a precisely defined nucleotide sequence). By "miRNA delivery" one can mean the delivery of miRNA mimetics/mimics to replenish the level of missing miRNAs in cells (the opposite effect is provided by antimiRs).

To improve the precision of our language, we changed miRNA to artificial miRNA (amiRNA) in the text.

8. It should be mentioned in the introduction that "spatiotemporal control of RNAi" can also be controlled by the use of cell type-specific promoters or the viral vector serotype used, etc.,

To address this point, we have added text regarding tissue specific promoter and AAV serotypes to the Conclusion section: "ORIENTRs also provide more cell-specific RNA interference that is

regulated by RNA profile to potentially reduce side effects on off-target cells. Currently researchers have been using tissue-specific promoters⁹ and AAV serotypes^{10,11} to confine RNAi to desired target tissues. However, their availability and specificity are still limited for applications in certain tissues. For instance, they lack specificity in recognizing different types of muscles¹² or nerves¹³, or distinguishing healthy and diseased cells. However, ORIENTRs can utilize transcriptomic differences as activation signals that can vary between cell types, along different cell developmental stages, in response to environmental stimuli, or when affected by disease. These transcriptomic differences can provide more accuracy and dynamic regulation to enhance the specificity of interference. Therefore, ORIENTRs could provide an added layer of regulation when used in combination with tissue-specific promoters and AAV delivery, and they could thus be an enabling tool for next-generation selective and safe AAV mediated RNAi gene therapy¹⁴⁻¹⁶”

9. The authors discuss the various possibilities of using the ORIENTR system in very general terms. In my opinion, the presented examples are not convincing and, most importantly, they are not supported by the results. It would be very valuable to demonstrate the usefulness of the ORIENTR system on a real example with a visible biological effect.

In the revised manuscript, we demonstrated that ORIENTR can sense mRNA expression in response to an environmental cue via the endogenous mRNA *Hsp70*, and produce a non-reporter amiRNA to knockdown the endogenous *KRAS* gene. Please see new subsection “Using ORIENTRs for sensing and knockdown of endogenous genes” and Fig. 4 in the revised manuscript. We believe that these demonstrations are salient examples showcasing the usefulness of ORIENTRs.

Minor

1. Page 2 - I guess the authors meant “systemic” instead of “systematic delivery”

We thank the reviewer for noting this error and have corrected the text.

2. Page 4 – „a viral miRNA sequence”, more specifically, it is an artificial miRNA targeting viral HSUR4

To clarify, miRNA-HSUR4-5p is a naturally existing miRNA that is derived from a primary HSUR4 transcript in HVS-infected cells, but it is not a microRNA targeting HSUR4¹⁷. We added this reference to the main text for clarification.

3. Page 5, line 5, line 6: „pri-miRNA-16-2” instead of „pri-miRNA” (these observations are not of a general nature but concern a specific case - pri-miR-16-2)

We have corrected this issue. All instances of pri-miRNA were changed to pri-miR-16-2.

4. Page 5 – „the scaffold lost its capacity to produce miRNA”- this is only a suspicion as the authors did not analyze processing efficiency, guide/passenger strand ratio, etc.

We changed the statement to “the scaffold failed to knockdown reporter GFP”.

We added Fig. S1f as supporting information for our hypothesis: “Our hypothesis that a functional pri-miR-16-2 scaffold is dependent on the basal stem structure without sequence

constraints was further confirmed by northern blot (Fig. S1f).” In this figure, the structure preserved scaffold produces amiRNA while the basal stem disrupted one did not produce any amiRNA that can be probed by northern blot.

5. The "Materials and methods" section does not contain important information, e.g. regarding the Northern Blot method; Luciferase assay – no information on the concentration of plasmids used.

We have updated the Methods section to provide this information.

6. Fig.2b- What is the percentage improvement in "leakage" of the proposed system compared to the original ORIENTR_2?

The reporter signal in presence of ORIENTR_2-4 increased by 34% compared with reporter signal in presence of original ORIENTR_2. We added this data in the revised manuscript: “Compared with the original construct, reporter signal in the presence of ORIENTR_2-4 increased by 34% while preserving responsiveness against its trigger (Fig. 2b).”

7. Fig. 2d – the graph does not show a significant improvement in "leakage" after adding the mentioned motif compared to the original molecule

We thank the reviewer for noting this issue. We agree that not all of the systems shown in Fig. 2d show decreases in leakage when the leak-reduction motif is added; however, the motif does on average increase the difference in signals obtained in conditions with and without the cognate trigger. Since the effect of this motif varies depending on the ORIENTR, we have chosen to update its name to the 3' stem loop instead. We have also updated the section discussing this motif in the revised manuscript.

8. Based on the results obtained, can a general principle for creating working pairs of trigger RNA and pri-miRNA be created?

We added the general design optimization guidance for improving working trigger-ORIENTR pairs in the Conclusion section: “When generating an ORIENTR for a particular RNA trigger molecule, our experience reveals several general principles for ORIENTR design. A stronger sequestration arm helps to decrease leaky amiRNA expression, while a longer toehold in the sensing region can facilitate RNA trigger recognition and interaction. Moreover, the trigger RNA should possess low secondary structure to facilitate ORIENTR recognition and ideally it should be present at a high copy number in the nucleus. Incorporating these principles should maximize the range of triggers and target genes to which ORIENTRs can be applied.”

9. error in the reference - in the conclusions, the last paragraph is 3738 instead of - 37, 38; similar, in the first paragraph of the first chapter, 2825, instead of 25, 28.

This error has been corrected.

10. Fig. S3 – no red arrows mentioned

Red arrows have been added to the figure.

Reviewer #3 (Remarks to the Author):

The authors report a conditional RNAi system that they call Orthogonal RNA Interference induced by Trigger RNA (ORIENTR), that provides a 14-fold increases in miRNA biogenesis upon activation in orthogonal libraries. They show that coupling ORIENTR triggers with Cas13d enhances dynamic range up to 31-fold. They report ORIENTRs that can direct a tissue-specific signature RNA to a targeted RNA interference response. They claim that this is the first demonstration of applying programmable cellular RNAs to regulate RNA interference in living cells. As far as I know, this is correct. They claim that this represents an advance toward introducing synthetic regulatory links into mammalian cells and organisms. I agree. Overall this paper makes a major step forward and is suitable for publishing in Nature Comm.

We thank Reviewer #3 for the positive appraisal of our work.

References

1. Partin, A. C. *et al.* Cryo-EM Structures of Human Drosha and DGCR8 in Complex with Primary MicroRNA. *Mol Cell* **78**, (2020).
2. Xie, J. *et al.* Effective and Accurate Gene Silencing by a Recombinant AAV-Compatible MicroRNA Scaffold. (2020) doi:10.1016/j.ymthe.2019.11.018.
3. Fellmann, C. *et al.* An optimized microRNA backbone for effective single-copy RNAi. *Cell Rep* **5**, (2013).
4. Huang, V. & Li, L. C. miRNA goes nuclear. *RNA Biology* vol. 9 Preprint at <https://doi.org/10.4161/rna.19354> (2012).
5. Liu, H. *et al.* Nuclear functions of mammalian MicroRNAs in gene regulation, immunity and cancer. *Molecular Cancer* vol. 17 Preprint at <https://doi.org/10.1186/s12943-018-0765-5> (2018).
6. Grimm, D. *et al.* Fatality in mice due to oversaturation of cellular microRNA/short hairpin RNA pathways. *Nature* **441**, 537–541 (2006).
7. Medley, J. C., Panzade, G. & Zinovyeva, A. Y. microRNA strand selection: Unwinding the rules. *Wiley Interdisciplinary Reviews: RNA* vol. 12 Preprint at <https://doi.org/10.1002/wrna.1627> (2021).
8. Chan Baek, S. *et al.* Structural atlas of human primary microRNAs generated by SHAPE-MaP. (2024) doi:10.1016/j.molcel.2024.02.005.
9. Rao, M. K. & Wilkinson, M. F. Tissue-specific and cell type-specific RNA interference in vivo. *Nat Protoc* **1**, 1494–1501 (2006).
10. Naso, M. F., Tomkowicz, B., Iii, W. L. P. & Strohl, W. R. Adeno-Associated Virus (AAV) as a Vector for Gene Therapy. *BioDrugs* **31**, 317–334 (2017).
11. Issa, S. S., Shaimardanova, A. A., Solovyeva, V. V. & Rizvanov, A. A. Various AAV Serotypes and Their Applications in Gene Therapy: An Overview. *Cells* vol. 12 Preprint at <https://doi.org/10.3390/cells12050785> (2023).
12. Tabebordbar, M. *et al.* Directed evolution of a family of AAV capsid variants enabling potent muscle-directed gene delivery across species. *Cell* **184**, (2021).
13. Goertsen, D. *et al.* AAV capsid variants with brain-wide transgene expression and decreased liver targeting after intravenous delivery in mouse and marmoset. *Nat Neurosci* **25**, (2022).
14. Harper, S. Q. Progress and Challenges in RNA Interference Therapy for Huntington Disease. *Arch Neurol* **66**, 933–938 (2009).

15. Hocquemiller, M., Giersch, L., Audrain, M., Parker, S. & Cartier, N. Adeno-Associated Virus-Based Gene Therapy for CNS Diseases. *Hum Gene Ther* **27**, 478 (2016).
16. Wang, W. *et al.* Efficient and Precise Processing of the Optimized Primary Artificial MicroRNA in a Huntingtin-Lowering Adeno-Associated Viral Gene Therapy in Vitro and in Mice and Nonhuman Primates. *Hum Gene Ther* **33**, 37–60 (2022).
17. Cazalla, D., Xie, M. & Steitz, J. A. A Primate Herpesvirus Uses the Integrator Complex to Generate Viral MicroRNAs. *Mol Cell* **43**, (2011).

REVIEWERS' COMMENTS

Reviewer #1 (Remarks to the Author):

In the revised version of their manuscript, the authors have addressed all points that I had raised. I am satisfied with the response to my comments.

A programable RNAi tool is certainly innovative and useful. Whether a miRNA can easily be replaced by many other siRNA sequences without losing Drosha activity needs to be tested case-by-case, I guess but would be doable. The method itself is also rather complex and already existing tools might be superior. However, it could be useful for specific applications such as programable inactivation of a gene in synthetic biology.